# Minimal Residual Disease in Acute Lymphoblastic Leukemia: Current Practice and Future Directions

**DOI:** 10.3390/cancers13081847

**Published:** 2021-04-13

**Authors:** Gloria Paz Contreras Yametti, Talia H. Ostrow, Sylwia Jasinski, Elizabeth A. Raetz, William L. Carroll, Nikki A. Evensen

**Affiliations:** 1Division of Pediatric Hematology Oncology, NYU Langone Health, New York, NY 10016, USA; gloria.contrerasyametti@nyulangone.org (G.P.C.Y.); Sylwia.jasinski@nyulangone.org (S.J.); elizabeth.raetz@nyulagone.org (E.A.R.); 2Department of Pediatric and Pathology, Perlmutter Cancer Center, NYU Langone Health, Smillow 1211, 560 First Avenue, New York, NY 10016, USA; Talia.ostrow@nyulangone.org (T.H.O.); Nikki.evensen@nyulangone.org (N.A.E.)

**Keywords:** acute lymphoblastic leukemia, minimal residual disease, flow cytometry, PCR, ddPCR, next generation sequencing, clinical significance

## Abstract

**Simple Summary:**

Acute lymphoblastic leukemia minimal residual disease (MRD) refers to the presence of residual leukemia cells following the achievement of complete remission, but below the limit of detection using conventional morphologic assessment. Up to two thirds of children may have MRD detectable after induction therapy depending on the biological subtype and method of detection. Patients with detectable MRD have an increased likelihood of relapse. A rapid reduction of MRD reveals leukemia sensitivity to therapy and under this premise, MRD has emerged as the strongest independent predictor of individual patient outcome and is crucial for risk stratification. However, it is a poor surrogate for treatment effect on long term outcome at the trial level, with impending need of randomized trials to prove efficacy of MRD-adapted interventions.

**Abstract:**

Acute lymphoblastic leukemia (ALL) is the most common pediatric cancer and advances in its clinical and laboratory biology have grown exponentially over the last few decades. Treatment outcome has improved steadily with over 90% of patients surviving 5 years from initial diagnosis. This success can be attributed in part to the development of a risk stratification approach to identify those subsets of patients with an outstanding outcome that might qualify for a reduction in therapy associated with fewer short and long term side effects. Likewise, recognition of patients with an inferior prognosis allows for augmentation of therapy, which has been shown to improve outcome. Among the clinical and biological variables known to impact prognosis, the kinetics of the reduction in tumor burden during initial therapy has emerged as the most important prognostic variable. Specifically, various methods have been used to detect minimal residual disease (MRD) with flow cytometric and molecular detection of antigen receptor gene rearrangements being the most common. However, many questions remain as to the optimal timing of these assays, their sensitivity, integration with other variables and role in treatment allocation of various ALL subgroups. Importantly, the emergence of next generation sequencing assays is likely to broaden the use of these assays to track disease evolution. This review will discuss the biological basis for utilizing MRD in risk assessment, the technical approaches and limitations of MRD detection and its emerging applications.

## 1. Introduction

Acute lymphoblastic leukemia (ALL) is the most common pediatric malignancy. In the United States, its incidence is 1.4 cases per 100,000 people [1]. Treatment outcome has improved significantly in the last 40 years as a result of treatment intensification, central nervous system (CNS) prophylaxis and the development of risk stratification allowing more customized treatments. Many clinical and biological variables are associated with treatment outcome with the principal ones being age, white blood cell count, the presence of CNS involvement at diagnosis, blast genotype and initial response to therapy as measured by the kinetics of disease regression. These variables are used in a variety of algorithms to predict the risk of relapse. Based on the predicted risk of relapse at diagnosis, low and intermediate risk patients in developed countries have a 5-year event free survival (EFS) rate of 90% and patients with high risk features can attain an 80% survival with augmented therapy [2,3]. In spite of this success, certain subgroups such as infants, adolescents and young adults and patients who relapse have an inferior outcome, making ALL one of the principal causes of pediatric cancer related death.

Since the early 1970s, several tumor growth kinetic and response models have informed core principles of tumor biology. Based on these fundamentals principles, ALL treatment backbones were optimized by alternating the maximal tolerated doses of non-cross-resistant drugs to reduce the tumor burden as rapidly as possible in order to eradicate drug resistant cells [4]. The overwhelming majority of patients achieve morphological remission (<5% marrow blasts), but that threshold fails to appreciate the substantial burden that may be remaining, of up to 10 billion blasts (10^10^) [5]. More sensitive and quantitative measurements are now better at discriminating levels of residual tumor burden with currently available assays measuring minimal residual disease (MRD) at a maximal sensitivity detection capacity of one blast in a background of 1 million cells.

## 2. Methods Of Minimal Residual Disease Detection

Numerous techniques are available for the detection of leukemic cells, however, for MRD detection, these methods must be specific, extremely sensitive, reproducible and broadly applicable. Each technique relies on identifying a different cellular target unique to the blast population, such as DNA (immunoglobulin heavy chain (IgH)/T cell receptor (TCR) gene rearrangement), RNA (fusion proteins) or surface proteins (cluster of differentiation (CD) markers). Each of them has unique advantages and disadvantages (Table 1). In order to assure accurate and reproducible MRD detection for clinical application across different laboratories, there are several groups dedicated to the standardization of protocols among institutions, such as the EuroFlow Consortium and the European Study Group on MRD detection in ALL (ESG-MRD-ALL).

### 2.1. Multiparametric Flow Cytometry for Surface Proteins

Multiparametric flow cytometry (MPFC) is used to measure MRD levels by identifying remaining leukemic cells based on surface protein expression (immunophenotyping). For flow cytometric analysis, samples are incubated with fluorochromes conjugated to antibodies specific to proteins of interest. Single cells flow past multiple lasers that excite each fluorochrome and the emitted fluorescence intensity is captured and converted into digital signals that can be analyzed by a computer (Figure 1A). Antibody and color (fluorochromes) combinations need to be optimized to ensure specificity and clear separation of all individual colors with limited overlap in emission spectra is required. For MRD purposes, four to twelve colors are commonly used [6,7]. Immunophenotyping is based on the expression of a combination of various “cluster of differentiation” (CDs) antigens that define specific cell lineages and state of maturation. In T-ALL, the presence of immature T cells found outside of the thymus are indications of MRD, and so regular precursor cell markers can be used for identification of samples taken from the bone marrow. In B-ALL, markers that distinguish blasts from normal precursor cells are required as they are both found in the marrow. Using flow, these cells are identified as having atypical surface antigens, combinations of antigens or quantities of specific antigens [8]. At diagnosis, flow cytometry can be used to establish specific antigen profiles for each individual patient, or leukemia-associated immunophenotypes (LAIPs). These LAIPs can then be used after induction to look for remaining disease in a case specific manner. However, LAIPs and normal cell populations can change over the course of treatment [9,10]. A second approach is to identify leukemic cells by how they deviate from normal hematopoietic populations using a defined set of antigens as opposed to using the diagnostic sample to determine the antigens of interest [11,12]. This method requires a sophisticated understanding of antigen expression throughout different lineages and maturation states in order to distinguish cell types. The patient specific and universal standardized approach can be used together to optimize blast identification and quantification [13].

Newer flow cytometry protocols are being developed by the EuroFlow Consortium, referred to by some as next generation flow (NGF) [14], to create highly standardized approaches for MRD detection [15]. First, a bulk lysis procedure was utilized to remove erythrocytes and concentrate leukocytes to increase the amount of cells that could be labeled. Second, multiple rounds of optimization phases were carried out to determine the most comprehensive 8-color panel. Two tubes are used with six identical backbone markers and two additional distinct markers in each tube, that further confirms the identity of low frequency cell types (Figure 1A) [14,15]. While four-color flow has a sensitivity of 0.01%, these newer approaches can achieve a sensitivity of 0.001% due to the ability to label >4 × 10^6^ cells [15,16]. Advancements in analytical tools to automate gating (Figure 1A) of distinct cell populations will help standardize these assays even further.

Overall, flow cytometric methods are applicable to >95% of patients and provide information about the immunophenotypic heterogeneity of leukemia and the cellular status of the bone marrow microenvironment that other methods of MRD lack [12]. While the cost is less and turnaround time is much faster for flow (1 day) in comparison to other methods described below (4 days to weeks), flow requires immediate processing as it is performed on live cells. It also requires a large number of cells to reach sensitivity levels beyond 10^−5^.

### 2.2. Polymerase Chain Reaction (PCR) Amplification of Antigen Receptor Rearrangements

Immunoglobulin and T-cell receptor antigen binding regions are made up of V (variable), D (diversity, IgH and TCRβ) and J (joining) segments. During lymphocyte development, the DNA encoding these genes undergo V(D)J rearrangement, forming unique combinations and highly variable junctional regions in each cell in order to increase the repertoire of antigen specificity (Figure 1B). These combinations (Figure 1B, black boxes) can serve as a clonal “fingerprint” allowing for each lymphocyte to be identified by its individual recombination [17]. Leukemic blasts are thought to originate from a single B/T cell clone, making their V(D)J repertoire monoclonal, allowing for the detection of the leukemic-specific rearrangements. Early methods of blast detection took advantage of V(D)J rearrangement by using southern blot analysis and later PCR amplification followed by polyacrylamide gel electrophoresis (PAGE) or capillary electrophoresis (CE) to detect the clonal leukemic V(D)J DNA.

Southern blotting relied on restriction digestion of DNA followed by hybridization with a combination of probes that could detect various V(D)J gene rearrangements. Based on a limited sensitivity range of 5–10% and the requirement for a large amount of starting DNA material, southern blotting was not an efficient method for MRD detection [18]. PCR based detection methods were designed based on the same concept that amplification of the specific IgH/TCR rearrangements in the leukemic cells would be visually detected utilizing CE or PAGE(Figure 1B). These methods, while requiring high levels of monoclonal blasts, need a lower amount of DNA and their sensitivity for detecting clonality ranged from 0.2–1% for PAGE down to 0.03–0.05% for CE [19].

In order to track specific leukemic clones throughout disease progression for the purpose of MRD and to increase sensitivity, IgH/TCR rearrangements are amplified using primers that anneal to conserved V-family framework regions and a J-family reverse primer at diagnosis (Figure 1B, black half arrows). The amplified product is then sequenced to determine patient specific rearrangements including the unique junctional regions between V-D and D-J segments (Figure 1B, black boxes). Patient specific primers (Figure 1B, dark green half arrow) are then used to amplify the target regions followed by electrophoresis [20,21]. Additional sensitivity can be achieved by using a double amplification approach, or nested-PCR approach in which the product from the first PCR is amplified using a patient specific primer (Figure 1B, dark green half arrow) [22]. Using these earlier methods, MRD was measured semiquantitatively as end point readings based on limited dilutions of the diagnostic sample (Figure 1B, PAGE). Numerous studies were able to detect leukemic cells down to one blast in a background of up to 10^6^ cells although stratification is usually based on the detection of one cell in >10^3^ cells following induction therapy using these methods [23,24,25].

Real-time quantitative PCR (RQ-PCR) is a more quantitative approach for tracking IgH/TCR gene rearrangements for MRD detection that measures fluorescent signals generated after each cycle of the PCR. As with conventional PCR, the leukemic specific IgH/TCR sequence needs to be determined from the diagnostic sample to design sequence-specific primers and fluorescent probes for each patient (Figure 1B, dark green half arrow and lime green line). The fluorescent signal increases exponentially as the amount of PCR products increase. Upon completion of all the PCR cycles, a background fluorescent signal is determined and the cycle at which samples rise above that level (cycle threshold (C_T_)) is indicative of the amount of initial starting material. The quantification of blasts post-induction relies on comparison to the limit of sensitivity determined using the diagnostic sample, which is generally 10^−4^ (Figure 1B, RQ-PCR) [26]. Thus, these assays are limited based on the availability of diagnostic material and need to be optimized for each individual case. Furthermore, cases with low tumor burden pose a problem for RQ-PCR if the target amplification is outside the C_T_ range considered accurate and reproducible (quantitative range) yet still above background levels for that particular assay. Cases that present with MRD levels in this range are considered “positive non-quantifiable” (PNQ). The European Study Group on MRD detection in ALL (ESG-MRD-ALL) has put forth guidelines for running, analyzing and interpreting MRD measurements based on RQ-PCR, making RQ-PCR for IgH/TCR rearrangements one of the most standardized methods for MRD detection for leukemia [27].

Digital-droplet-PCR (ddPCR) technology has made it possible to determine the absolute amount of specific DNA molecules in a given sample without the need to compare to a standard curve. DNA targets are partitioned using water–oil emulsion technology into 20,000 individual droplets that end up with either one, a few or no target sequences. PCR amplification of the target region occurs within the droplets and the fluorescent intensity is measured in each droplet following completion of the reaction (Figure 1B, ddPCR). A fluorescent threshold is set, droplets are counted as either positive or negative, and the precise amount of target sequence is determined based on Poisson’s statistics [28]. The patient specific primers and probe used in ddPCR are designed the same way as those used for RQ-PCR (Figure 1B, dark green half arrow and lime green line). DdPCR eliminates the requirement for diagnostic material to establish a standard curve and reduces both non-specific target competition and the presence of inhibitors, which can reduce amplification efficiency due to the partitioning of reactions. However, it is more expensive and requires special equipment. It is also more effective at discriminating positive MRD within the range considered PNQ by RQ-PCR. Starza et al. directly compared ddPCR to RQ-PCR for IgH/TCR rearrangement detection in patients enrolled on the GIMEMA trial demonstrating 88% of samples were concordant and similar sensitivity levels (10^−4^) are able to be obtained without the use of a standard curve [29]. These assays were compared again, in a subset of patients enrolled on the AIEOP-BFM ALL 2000 trial. From this cohort, 45 slow early responders (day 33 high MRD and day 78 PNQ MRD by RQ-PCR) were reanalyzed utilizing ddPCR. This identified 13 patients with quantifiable ddPCR MRD at day 78 of therapy who had a 5-year EFS of 36% compared to 32 patients with neg/PNQ day 78 MRD by ddPCR with a 5-year EFS of 77%. The remainder of responder group outcomes were similar, demonstrating that ddPCR is as sensitive as RQ-PCR and can provide a potentially more accurate prognostic stratification for cases defined as PNQ MRD by RQ-PCR [30]. ddPCR has recently been used to track specific genetic leukemia-associated mutations in an effort to detect rising subclones during maintenance therapy using peripheral blood samples. *NT5C2* and *PRPS1* variants were detected down to a sensitivity of 0.008% 116 days and 0.005% 58 days before frank relapse, demonstrating the feasibility of utilizing ddPCR to detect rare cell populations throughout disease progression [31].

There are several advantages of tracking IgH/TCR rearrangements during disease progression. The rearrangements are blast specific and are the most broadly applicable target for MRD, as it can be used on >95% of patients. The sensitivity of detection is as low as 10^−4^–10^−5^. Some disadvantages include the requirement of diagnostic samples and patient specific primers/probes rather than universal tools. Although it was originally thought that blasts originated from a single dominate IgH/TCR clone, oligoclonal rearrangements do occur and can lead to false negatives if the wrong clone is tracked. Additionally, the V(D)J rearrangements can change in subclones through ongoing rearrangement, making the primers based on initial patient samples obsolete. Therefore, choosing multiple clones and using more stable D-J junctions can increase the stability of the targets and the accuracy of the assay. False positives can also occur if there is non-specific primer annealing resulting in amplification of DNA that is not the correct target [8].

### 2.3. PCR Amplification of Chromosomal Translocations

PCR based methods can also be used for chromosomal translocations that are detected at diagnosis using cytogenetic techniques such as fluorescence in situ hybridization (FISH). There are commonly recurring chromosomal translocations/gene fusions in ALL, including *EVT6/RUNX1* (20–25%), *E2A/PBX1* (4–8%), *MLL/AF4* (2–6%) and *BCR/ABL* (Philadelphia chromosome (Ph+) (2–4%) [32,33]. For detection of translocations, RNA is more commonly used as the substrate compared to DNA because the exact breakpoint/fusion site of a translocation occurs in different intronic regions and spans large segments of DNA making it difficult to determine for each patient. Furthermore, if the fusion intron regions span too large of an area, they are not suitable for detection via PCR [16]. In contrast, the spliced mRNA product (which excludes introns) that gives rise to the fusion protein is similar between patients, and shorter, thus allowing for universal primers (Figure 1C, black half arrows) [12,34]. The mRNA transcript is reverse transcribed into cDNA, which is then amplified by PCR using the universal primers (Figure 1C, black half arrows). In contrast to IgH/TCR RQ-PCR, detecting this fusion transcript using mRNA does not require patient specific assay development and optimization, making it a more time efficient and cheaper method of detection. These fusion transcripts also contribute to the oncogenic process, unlike the IgH/TCR rearrangements. However, RNA is less stable and is a more difficult substrate to work with. Since the number of mRNA transcripts can vary within the blasts, understanding the quantitative implications of detection is not straightforward [8]. Additionally, housekeeping genes need to be run for each assay in order to control for differing efficiencies in cDNA production with subsequent normalization of the relative number of transcripts [34]. Reverse transcription RQ-PCR of translocations has a sensitivity down to 10^−6^. Importantly, studies have found that in Ph+ patients, RQ-PCR of *BCR/ABL* fusion transcripts is more effective at earlier detection and is clinically relevant in predicting relapse [35].

### 2.4. Next Generation Sequencing

Next generation sequencing (NGS) is a newer method being applied to MRD detection. Similar to IgH/TCR identification required for RQ-PCR analysis of MRD, NGS requires that patient specific clonal V(D)J rearrangements be identified at diagnosis to allow for tracking of these clones throughout disease progression. However, NGS amplifies DNA by multiplexed PCR using a combination of universal primers containing molecular barcodes that amplify all areas of interest, including IgH gene segments and housekeeping genes (adaptive biotechnologies clonoSEQ assay) (Figure 1B, black half arrow with green or pink tail). The primers are designed against conserved regions within the V(D)J genetic regions, rather than patient specific junctional regions, to capture the entire region. Efforts are being made to standardize NGS assays for IgH/TCR MRD identification by the EuroClonality working group [36], potentially making it a more broadly applicable approach [13]. It also eliminates the time-consuming optimization steps required for each patient-specific assay design for RQ-PCR. NGS is able to identify most, if not all, possible rearrangements since each DNA molecule is sequenced, thus allowing for a more comprehensive understanding of the clonal repertoire of ALL and how it changes throughout disease progression. The sensitivity of this method depends on the amount of DNA used, and can range from 10^−4^ to 10^−7^ [16,37]. Importantly, NGS can also clarify the PNQ results from RQ-PCR due to the greater depth of sequencing information. A study measuring MRD post hematopoietic stem cell transplant (HSCT) found that most PNQ cases that were determined to be negative by NGS did not relapse without any changes in treatment, highlighting this difference [38]. NGS can also be performed using RNA as a substrate rather than DNA, referred to as RNAseq. A recent study has shown that using RNAseq to identify the IgH/TCR rearrangements in diagnostic samples can identify small subclones with different IgH rearrangements [39].

NGS can also be used for detection of chromosomal translocations. Targeted panels can be designed to capture specific DNA regions that are then amplified and sequenced. This allows for detection of multiple targets within one assay as compared to designing and running individual PCR reactions for each target. The depth of NGS provides more precise base-level breakpoint information for the translocations and can potentially identify unknown fusion partners [40].

While NGS cost and workflow time has decreased, it still requires sophisticated bioinformatic analysis that includes alignment to a reference genome, variant calling and finally filtering based on known variants and duplicated genomic regions. Since it is a relatively new method, a standardized NGS method still needs to be developed for wider use. Additionally, more studies are needed to evaluate the specificity and sensitivity of NGS of chromosomal translocations in order for it to be used as a standard method of MRD detection.

## 3. Clinical Relevance Of Minimal Residual Disease

The rate at which ALL regresses after initiation of induction therapy varies from patient to patient, and a number of studies have demonstrated it is the most robust and independent prognostic factor [41,42,43,44]. A consensus regarding the best threshold and time point(s) to evaluate the response to induction has been refined across different groups in the last few decades. In the early 2000s, the Children’s Oncology Group (COG), St. Jude Children’s Research Hospital (SJCRH) and the AEIOP-BFM defined the end of induction (EOI) as the most appropriate initial time point to evaluate MRD for risk-stratified therapy [44,45,46,47,48,49,50,51].

Overall, patients with higher residual disease following treatment have a worse outcome. The COG first measured the impact of different flow cytometry MRD levels on EFS at the EOI. In a cohort of 1971 patients enrolled on COG-P9904 (NCT00005585), COG-P9905 (NCT00005596) and COG-P9906 (NCT00005603) trials. The MRD response was classified in three groups >1%, 0.01–1% and <0.01%. Those with <0.01% MRD had an EFS of 88% ± 1%, in contrast to an EFS below 60% for all other groups. This suggested that utilizing flow cytometry based EOI MRD threshold of <0.01% (10^−4^) is an appropriate variable to identify patients at an increased risk of relapse [44]. Unfortunately, two thirds of relapses originate from EOI MRD negative patients. To better dissect the impact of MRD kinetics on outcome, the COG measured MRD at an earlier time point, namely day 8 peripheral blood status by flow cytometry. Patients with day 8 MRD > 1% had inferior outcomes compared to those with MRD ≤1% even if they cleared MRD by EOI (day 8 MRD positive 5-year EFS of 79% ± 4%; vs. 90% ± 1%, if MRD-negative, *p* < 0.001) [44]. The AIEOP-BFM ALL 2000 trial (NCT00430118 for BFM and NCT00613457 for AIEOP) evaluated the prognostic impact of molecular MRD at two different time points, Day 33 and Day 78 (equivalent to EOI and end of consolidation across different protocols worldwide) and classified B-ALL patients (*n* = 3184) by MRD status: MRD-standard risk if negative at a level of <10^−3^ at the first time point, MRD-intermediate risk if positive at the first time point but negative at the second time point and MRD-high risk if persistent disease at both time points. They found that MRD significantly correlated with outcome with 5-year EFS of >90%, 78% and 50% in the standard, intermediate and high risk groups, respectively, *p* < 0.001 [45].

In a later COG analysis, 7430 children with the National Cancer Institute (NCI) standard and high risk B-ALL were randomized to augmented therapy based on a combination of morphological response and MRD measured by MPFC at the EOI. MRD < 0.01% was again an independent predictor of outcome (5-year disease free survival [DFS] of 89% versus 72%, respectively *p* < 0.001). Furthermore, a better 5-year DFS was noted in a subset of patients with 0.01–0.1% MRD treated with augmented Berlin–Frankfurt–Munster therapy (ABFM) with two interim maintenance and delayed intensification phases, suggesting intensification based on response rescues some poor risk patients (5-year EFS 90% ± 6% with ABFM vs. 77% ± 3% without ABFM). This data validated a 0.01% threshold for intensification of therapy at the EOI [47]. Investigators at SJCR also investigated the impact of MRD thresholds in a cohort of 455 children with B-ALL measuring MRD utilizing RQ-PCR of antigen–receptor genes. They specifically looked at the cumulative risk of relapse (CRR) of patients otherwise considered MRD negative using the 0.01% threshold, and demonstrated that a persistent low level disease of 0.001% to <0.01% was associated with a CRR of 12.7% compared with 5.0% for those with undetectable MRD (<0.001%) (*p* < 0.047) [48].The association of MRD status with outcome has been confirmed over time in both pediatric and adult disease. Berry et al. performed a meta-analysis with 13,637 pediatric and adult patients. They analyzed the correlation of MRD-based disease response with 10-year EFS. To compare MRD methods (flow or PCR), threshold (<0.01%) and time point (EOI or consolidation) across the studies, Bayesian hierarchical analysis was performed to calculate the hazard ratio associated with positive or negative MRD status. Overall, pediatric patients had better survival rates than adults and both pediatric and adult MRD negative patients were more likely to survive (pediatric 10-year EFS 77% vs. 32% if MRD negative vs. positive; adult 10-year EFS 64% vs. 21%, if MRD negative vs. positive) [49].

Wood et al. utilized COG samples from 579 patients with B-ALL to study the sensitivity of MPFC in contrast to NGS. Their results validated <0.01% as the appropriate threshold to define MRD positivity at the EOI in both standard and high risk patients. NGS and flow cytometry both demonstrated a strong correlation with outcome (5-year EFS *p* = 0.009, 95% confidence interval (CI) 1.057, 1.471 and overall survival (OS) *p* = 0.074, 95% CI 0.976, 1.627). NGS has higher sensitivity than MPFC with a capability of identifying <0.001%—<0.0001% leukemic blasts (Figure 1B). Within the standard risk population, an MRD status of <0.0001% by NGS identified a unique group of patients with 8-year EFS of 98% that may be candidates for therapy reduction. Interestingly, this outstanding outcome was not observed among high risk patients with NGS MRD < 0.0001%, who had a 5-year EFS 92.7%. This illustrates that MRD must be interpreted within the context of clinical and biological disease features [50,51].

## 4. MRD Status Interaction with Other Genetic High Risk Groups

Several structural and numerical chromosomal abnormalities offer insight into the biology of ALL and are prognostically relevant. A central question is the relevance of genetic subtype in risk stratification if MRD is used. MRD response differs among different genetic subtypes demonstrating that the underlying disease biology is the key driver of treatment response. Patients who have favorable cytogenetics (e.g., *ETV6/RUNX1)* or neutral cytogenetics (e.g., *TCF3/PBX1*), which are particularly known to have a rapid treatment response and outstanding survival, are more likely to have salvageable disease when they relapse [52].

O’Connor et al. [53] evaluated MRD from a different perspective in order to integrate the value of disease biology provided by genetic subtypes. They evaluated MRD as a continuous variable utilizing IgH/TCR gene PCR and integrated it with the genetic subtype. The risk of relapse was strongly associated with MRD in each genetic subgroup. However the relapse risk associated with a single MRD value varied significantly between genetic subgroups. The results suggest that a single threshold may not be appropriate across all subgroups. This analysis has informed individualized thresholds for good, intermediate and high risk cytogenetics in the current UK and COG clinical trials [51,54].

### 4.1. Infant B-ALL

Infant ALL is a biologically distinct entity with an overall poor outcome. It is characterized by the presence of *KMT2A* gene rearrangements at 11q23.3 with one of numerous fusion partners, a high white blood cell count upon presentation and an immature B cell progenitor immunophenotype (CD10^−^, CD19^+^ and HLA^−^DR^+^). Both IgH/TCR rearrangements and *KMT2A* rearrangement are targets for PCR based MRD measurement. Given the frequency of oligoclonal IgH/TCR rearrangements, *KMT2A* fusion targeting is a more accurate approach. In addition, normal bone marrow lymphoid progenitors cells (CD19^+^, CD10^+^ and/or CD34^+^) are extremely sensitive to induction therapy, and cells detected during B-ALL therapy with this immunophenotype are assumed to be leukemic. Therefore, a simplified flow cytometry based MRD analysis faces the challenge of false negative results in infant B-ALL, making PCR or a sophisticated flow cytometry analysis the preferred methods for MRD detection in this subset of patients [54].

Van der Velden et al. studied the prognostic significance of RQ-PCR based MRD targeting IgH/TCR and *KMT2A* rearrangements in infant ALL in 99 patients treated in the Interfant 99 trial (NCT00015873). After induction therapy, 40% of the patients had detectable MRD (high (10^−3^–10^−2^) or very high (>10^−2^) MRD levels) and 50% became negative (below the quantitative range of at least 10^−4^) by the end of consolidation. It was also noted that *KMT2A* rearranged patients had a significantly slower response than infants with germline *KMT2A* rearrangement. Important conclusions of the study include: (1) end of consolidation (EOC) MRD dissects disease response patient distribution into two outcome groups, one with long term remissions (MRD < 10^−4^), and another one with extremely poor prognosis (MRD > 10^−4^) but in general relapses occur early making EOC an inadequate time point to change management, (2) EOI MRD significantly discriminates outcome with a wide range of DFS values, indicating that this time point is appropriate for risk stratification and (3) incorporating MRD status in the clinical based risk stratification (WBC >300 × 10^9^/µL, age < 60 months and presence of *KMT2A* rearrangement) allows for easy identification of high risk subsets eligible for novel treatment approaches in first remission [55].

### 4.2. Philadelphia Chromosome Positive B-ALL

Philadelphia chromosome positive (Ph+) ALL occurs in about 2–3% of children with ALL. Historically, these patients fared poorly despite efforts to augment therapy. However, after the introduction of tyrosine kinase inhibitors (TKI), their outcomes have improved considerably. MRD monitoring in this subset has been studied by targeting the DNA junctional sequences of IgH/TCR and/or *BCR/ABL1* fusion through PCR, or by flow cytometry. Cazzaniga et al. recently demonstrated that MRD monitoring using both targets is appropriate to assess response and predict outcome, with a correlation of 69% between methods. In this cohort, *BCR/ABL1* MRD levels were higher than those based on IgH/TCR at any time point [56]. A potential biological explanation for this discrepancy is that the *BCR/ABL1* translocation occurred in a multipotent progenitor cell, leading to multilineage clone involvement with non-leukemic cells being detected by the assay [57].

Despite the rarity of this genotype, COG and the European Intergroup have reported the revolutionary impact of TKIs in the standard treatment of Ph+ B-ALL in three clinical trials (EsPhALL [NCT00287105], AALL0031 [NCT00022737] and AALL0622 [NCT00720109]) [56,58,59]. These studies have similar data on MRD kinetics, and demonstrated overall early achievement of MRD negativity lowers the cumulative incidence of relapse (CIR). For MRD negative patients, EFS rates were 60% for AALL0622 (5 years), 61% for EsPhALL (4 years) and 58% overall for AALL0031 (5 years) [60,61,62]. AALL0622 incorporated dasatinib as opposed to imatinib early in the conventional chemotherapy backbone (3 weeks). The authors hypothesized that the enhanced CNS penetrance of dasatinib could replace CNS radiation to prevent relapse. In this cohort, discontinuous dasatinib led to an improved MRD response (59% of AALL0622 patients had EOI MRD < 0.01% vs. 25% on AALL0031 (*p* = 0.001); 89% of AALL0622 patients had EOC MRD < 0.01% compared to 71% on AALL0031 (*p* = 0.03)). Strikingly, an improved early response did not translate to improved EFS. The authors attributed this observation to an increase in isolated CNS relapses of 15% compared to 5% with imatinib on COG AALL0031. Additionally, patients with *IKZF1* deletion had decreased EFS compared to those with wild type *IKZF1* despite early MRD negativity (5-year OS *IKZF1* deletion 80% vs. wild type 100% (*p* = 0.04); and EFS *IKZF1* deletion 52% vs. wild type 82% (*p* = 0.04)). This data demonstrate that the subset of Ph+ ALL patients with high risk genetic markers, such as *IKZF1* alterations and EOI MRD positivity, have more aggressive disease that may benefit from hematopoietic stem cell transplant (HSCT) [60]. In a single center study (*n* = 68), hematologic response after induction and *BCR/ABL* levels at three months were shown to predict relapse and define a high risk group with improved long term EFS if treated with HSCT [63]. Current and future trials must assess the incorporation of MRD status in concert with other biological variables when defining high risk patients who are allocated to alternative therapy.

### 4.3. Philadelphia (Ph)-Like B-ALL

Ph-like B-ALL is a subgroup that expresses the *BCR/ABL* gene expression profile in the absence of the *BCR/ABL* fusion. This subset is clinically recognized as a high risk group with poor outcomes, with a 5-year EFS of approximately 60% compared to 85% of other non Ph-like high risk groups [64,65,66,67,68]. These patients can be further grouped into 4 clusters: *CRLF-2*, *ABL –class* translocations, *EPO* receptor and *JAK2* translocations and JAK/STAT or RAS mutations. This group has consistently demonstrated a high probability of MRD positivity after induction compared to other high risk groups (63.8% vs 19.7%, *p* < 0.0001, reported by COG) and inferior EFS (5-year EFS 55% if EOI MRD positive vs. 73% if EOI MRD negative) [68]. Thus, MRD positive patients are good candidates for immunotherapy and/or HSCT. Given the strong association of MRD positivity with relapse after HSCT, every effort to augment the high risk conventional chemotherapy backbone should be done to eradicate disease prior to transplantation.

### 4.4. Hypodiploid B-ALL

Hypodiploid B-ALL (<45 chromosomes) is another group with a very poor prognosis with an OS of 50% [62]. Patients with near-haploidy (24–31 chromosomes) and low-hypodiploidy (32–39 chromosomes) ALL fare particularly poorly. Mullighan et al. first demonstrated in a small cohort of 20 patients that MRD is the most significant prognostic indicator in hypodiploid ALL and that MRD directed therapy can improve outcome [69]. These results were further expanded and validated by the COG that reported the retrospective outcome of 131 patients with hypodiploid ALL, initially enrolled on COG AALL03B1 (NCT00482352). This study demonstrated the poor outcome of EOI MRD positive patients (EOI MRD ≥ 0.01% had 5-year EFS and OS of 26.7% ± 9.3% and 29.3% ± 10.1%, respectively) and HSCT had no significant impact on outcomes [70]. Similar results were yield in a large multinational retrospective review lead by Pui et al. [71]. In this study of 272 hypodiploid ALL patients across 16 cooperative study groups, the overall survival was 50%, consistent with prior reports. However, an exception was observed in two subsets, namely patients with high hypodiploidy (40—<44 chromosomes) had a 5-year EFS of 74% and EOI MRD negative patients had a 5-year EFS of 75%. After excluding these groups, there was no difference in outcome among patients treated with chemotherapy alone vs. HSCT when therapy was directed by MRD status (53.0% vs. 59.8%, *p* = 0.47) [71].

### 4.5. T-Cell Acute Lymphoblastic Leukemia (T-ALL)

T cell acute lymphoblastic leukemia represents 15% of new ALL diagnoses. It is biologically distinct from B-ALL with outcomes that have now reached those in B-ALL with refined treatment approaches [72,73,74,75,76,77,78]. The clinical variables used to risk stratify B-ALL do not have the same prognostic value in T-ALL. MRD remains a key independent prognostic factor, however, the kinetics of blast regression are slower.

A subset of 464 patients with T-ALL on the AIEOP-BFM ALL 2000 trial (NCT00430118 for BFM and NCT00613457 for AIEOP) underwent the same induction and MRD risk based stratification as their B-ALL counterparts (standard risk if negative at a level of less than 10^−3^ at the first time point, intermediate risk if negative at a second time point and high risk if persistent disease at a second time point). Results differed significantly from the B-ALL population, demonstrating a slower kinetic pattern of blast regression. Only 16% of T-ALL patients cleared MRD at the EOI as opposed to 42% of B-ALL, and they were more likely to have persistent disease than B-ALL patients at EOC (20.9% vs 5.9%). A substantially better outcome was observed for T-ALL standard and intermediate risk groups compared to high risk (7-year EFS 91.1%, 80.6% and 49.8% respectively, *p* < 0.001). This analysis highlights the difference between these two biologically distinct entities and demonstrates EOC is a more suitable time point for T-ALL risk stratification [46].

T-ALL phenotype can also be classified according to the intrathymic differentiation state. Early T cell precursor (ETP) ALL is a subtype characterized by a unique immunophenotype (cCD3^+^, sCD3^−^, CD1a^−^, CD2^+^, CD5^dim^ (<75%^+^) and CD7^+^) and positivity for stem cell and/or myeloid markers that represents 15% of T-ALL cases and was previously associated with a poor outcome [79,80,81]. ETP ALL has been associated with a higher disease burden at the EOI with a higher proportion of induction failures. In COG AALL0434 (NCT00408005), among the 1144 patients with T-ALL, 11% had ETP ALL. This subset had an 8% induction failure rate compared to 1% in non-ETP ALL, and 81% were MRD positive (>0.01% by flow cytometry) at the EOI. However, with the current MRD based therapy, non-ETP ALL and ETP ALL outcomes were similar. Five year EFS and OS rates were 87% and 93% respectively for ETP ALL, compared to 86.9% and 92%, respectively for non-ETP ALL. Interestingly, there was no difference in the total cohort EFS or OS for Day 29 MRD < 0.01% vs. 0.01–1.0%, suggesting the worse outcome for EOI MRD > 1% [82]. Similar data was reported in the ETP cohort of patients enrolled in the Medical Research Council UKALL 2003 trial (NCT00222612). Here, the ETP subgroup (16% of T-ALL patients) had a non-significantly inferior 5-year EFS (76.7% vs. 84.6%, *p* = 0.2) and OS (82.4% vs. 90.9%, *p* = 0.1), and a higher relapse rate (18.6% vs. 9.6%, *p* = 0.1) compared to non-ETP T-ALL. The MRD impact in the ETP group could not be determined as the proportion of standard risk patients was too low to allow for a comparison. Given the current ETP-ALL outcome based on contemporary therapy, it may be considered an intermediate risk group, which does not warrant experimental treatment or first remission allogeneic HSCT for the group universally [74].

## 5. MRD Prognostic Value Peri-Stem Cell Transplant

Allogenic HSCT can be pursued for ALL with very high risk features in first remission and in cases of relapsed or refractory disease. A number of studies have described the prognostic significance of MRD in the peri-transplant period. Knechtli et al. in 1992 showed 2-year EFS of 0% for high MRD level by clonal band evident after electrophoresis (sensitivity 10^−2–^10^−3^), 36% for low-level positive MRD detected only after oligoprobing (sensitivity 10^−3^–10^−5^) and 73% if negative MRD prior transplant (*p* < 0.001) [83]. Bader et al. in 2009 showed a probability of EFS (pEFS) and CIR in 45 patients with RQ-PCR MRD >10^–4^ was 0.27 and 0.57 compared with 0.60 and 0.13 in 46 patients with MRD < 10^–4^ (EFS *p* = 0.004; CIR *p* = 0.001) [84]. Leung et al. in 2012 showed in 64 patients with ALL the 5-year OS was 49% if MRD was detectable (≥0.01% by flow cytometry) and 88% if it was not [85]. Umeda et al. in 2016 reported a small cohort of 33 ALL patients with complete remission, from which 3 patients had positive MRD > 0.01% by flow cytometry and relapsed within the first year post transplant, as opposed to the majority of the cohort that was MRD negative and had a 3-year CIR of 27.3% [86]. In a cohort of 56 patients with B-ALL enrolled on COG trial ASCT0431 (NCT00382109), Pulsipher et al. demonstrated NGS-MRD was highly predictive of relapse and survival as early as 30 days after HSCT (serial MRD at days +30, +100 or 8–12 months after HSCT). Post-HSCT NGS-MRD positive patients had an estimated relapse probability of 67% compared to 25% for post-HSCT NGS-MRD negative patients (*p* = 0.01) [87]. Serial MRD measurements after HSCT are critical for early identification of impending relapse and provide clinicians with a window to act on MRD in order to prevent overt clinical relapse. It is not within the scope of this review to address the potential strategies to improve this outcome, such as pretransplant intensification therapy or post-transplant immunomodulation treatment options, but to highlight pre and post-transplant MRD are the strongest predictors of HSCT outcome.

## 6. Current Recommendations and Dilemmas

As reviewed here, the ability to measure MRD has had a profound impact on ALL treatment and has emerged as the single most important variable in predicting outcome. Treatment stratification based on MRD has led to improvements in survival. Patients with slow kinetics of disease reduction receive augmented therapy thereby minimizing relapse, while patients with rapid clearance are spared the short and long term toxicity of additional treatment intensification. While incorporation of MRD has improved stratification, it does not completely replace other prognostic biomarkers, therefore using a combination of clinical and biological variables provides the most robust predictor of overall response. Moreover, it must be emphasized that up to 50% of relapses occur in patients predicted to have an excellent prognosis, so the search for additional predictive biomarkers is critical.

Many approaches to MRD testing have been developed with new methods with higher sensitivities (below 10^-6^) on the horizon. Whether MRD detected at this lower level is clinically significant is unknown, with the remaining questions including the optimal level of MRD and time point(s) that provide the best assessment of overall prognosis. Additional factors include local access to the technology with external validation of results while the assay is being introduced, cost (critical for resource challenged countries), lack of standardization across different laboratories and turnaround time that allows for rapid implementation of treatment decisions based on results.

Given the overall good prognosis of childhood ALL patients, clinical trials take many years to complete and thereby treatment advances are relatively slow. This is particularly frustrating to clinicians, parents and patients especially with new precision medicine and immunotherapy approaches now available. Thus, it is not surprising that many investigators questioned whether MRD can be used as a surrogate endpoint of outcome. A meta-analysis specifically designed to examine differences in survival between treatment interventions during induction and the ability of EOI MRD to predict outcome failed to show that MRD can be used as a surrogate for treatment effect on EFS at the trial level. This indicates the need to demonstrate the association between a therapeutic intervention and long-established treatment endpoints such as event free and overall survival [88]. However, MRD is used in the design of studies to identify patients for novel treatment interventions and has been used for regulatory accelerated approval of effective, novel regimens for patients with ALL such as the recent Food and Drug Administration approval of blinatumumab in 2018 for EOI MRD positive patients with B-ALL [89]. The current data advise oncologists, researchers and regulatory groups to continue to refine the application of MRD response approaches to complex ALL therapy but to carefully consider its limitations on predicting long term effects.

## Figures and Tables

**Figure 1 cancers-13-01847-f001:**
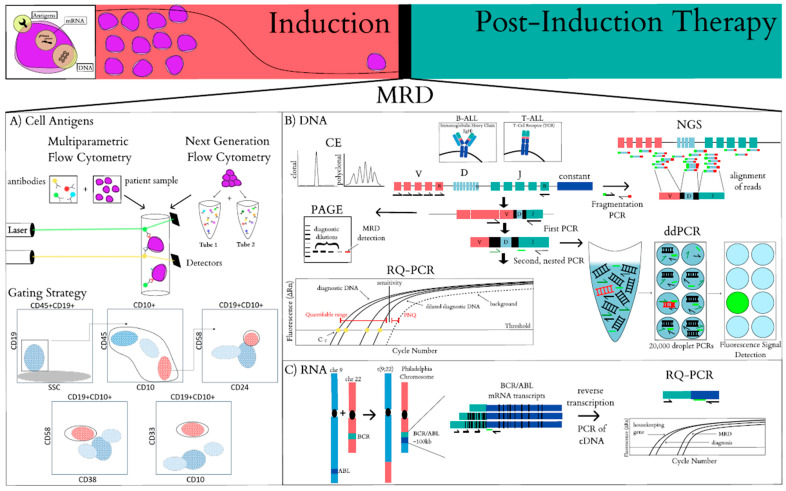
Methods of detection for MRD. (**A**): Representative schematic of multiparametric flow cytometry detection of the leukemia-associated immunophenotype. Antibodies conjugated to individual fluorochromes are incubated with the patient sample, either using 4-5 colors for traditional methods, or 8 colors (targeting 10 different antigens) in two separate tubes with next generation flow cytometry. Schematic example showing a gating strategy of a post induction sample analyzed with an informative antibody panel that allows the identification of residual leukemic blasts by deviation from normal lymphoid progenitors based on lineage and maturational stage. Examination of this CD45dimCD19+CD10+ population reveals blasts that are CD58+CD24+CD38dimCD33+. (**B**): MRD detection using DNA as a substrate for targeting the immunoglobulin heavy chain (IgH) for B-ALL or the T cell receptor (TCR) for T-ALL. Black boxes between V(D)J segments represent patient specific rearrangements formed during V(D)J recombination. Five representative schematics contrasting the assays utilizing this target, counterclockwise: capillary electrophoresis (CE), polyacrylamide gel electrophoresis (PAGE), real time quantitative polymerase chain reaction (RQ-PCR), digital droplet polymerase chain reaction (ddPCR) and next generation sequencing (NGS). (**C**): MRD detection using RNA as a substrate to track chromosomal translocations. An example is shown for the reciprocal translocation of t(9;22)(q34;q11) that results in the BCR/ABL oncogene.

**Table 1 cancers-13-01847-t001:** Comparison of technical methods for minimal residue disease (MRD) detection.

	Multiparametric Flow Cytometry	IgH/TCR RQ-PCR	Translocation RQ-PCR	Digital Droplet PCR	Next Generation Sequencing
**Requires patient** **specific design and optimization**	Yes for LAIP No for newer approaches	Yes	No	Yes	No
**Sensitivity**	4 color—10^−4^ 8 color—10^−5^ 8 colors with >4 × 10^6^ cells—10^−6^	10^−4^–10^−5^	10^−4^–10^−5^	10^−5^	10^−5^–10^−7^ Depending on amount of DNA
**Quantification**	Absolute	Semi	Semi	Absolute	Absolute
**Quantification based on diagnostic** **material**	No	Yes	No	Yes	No
**Applicability**	>90%	90–95%	30–40%	90–95%	>95%
**Turnaround time**	hours	weeks	days	hours	1 week
**Substrate** **-ease of use**	Cell suspension -unstable	DNA -stable	RNA -unstable	DNA -stable	DNA -stable
**Additional patient** **information**	Identifies heterogeneity of whole population	None	None	None	-Identifies clonal repertoire -Identifies precise breakpoints

Abbreviations: RQ-PCR, real-time quantitative PCR; LAIP, leukemia-associated immunophenotypes.

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
