# Peer review of "Minimal Residual Disease in Acute Lymphoblastic Leukemia: Current Practice and Future Directions"

_cancers, 2021, doi:10.3390/cancers13081847_

Round 1
Reviewer 1 Report
The authors Yametti et al . have written an important summary on the importance of diagnosing MRD in ALL. The review article should be published with slight improvements or additions.
- The different methods for calculating MRD (ct and others) should be evaluated in a separate paragraph regarding strengths and weaknesses (absolute versus relative quantification).
- The authors should mention the issue of standardisation regarding the performance of interlaboratory comparisons.
- Figure 1 needs to be revised. The areas of flow MRD and molecular genetic MRD should be shown separately. The legend is too long and needs to be shortened.
Reviewer 2 Report
The review is clearly organised and written; it provides good information on the topic with satisfying level of details and appropriate references. The only comment is that the text under figure 1 is very extensive and detailed, perhaps part of that text should go in the main body of the review instead.
Reviewer 3 Report
In the review entitled "Minimal residual disease in acute lymphoblastic leukemia: Current practice and future directions", the authors described the biological basis for utilizing MRD in risk assessment, considering also technical approaches and related limitations.
Major revisions:
In my opinion paragraphs 3-4-5-6 should be sub-paragraphs of nr.2, "Mehods Of Minimal Residual Disease Detection", and the amount of technical details should be reduced, supposing that the reader knows the basis of the presented methodologies.
Minor revisions:
- In the paragraph nr.3, "Multiparametric Flow Cytometry For Surface Proteins" line 75-85, flow cytometric analysis has been described with too much details.
- In the paragraph nr.4, "Polymerase Chain Reaction (PCR) Amplification of Antigen Receptor Rearrangements", in my opinion lines 124-129 have to be removed.
- The description of the legend of Figure 1 is redundant if compared to the main text. Insert the detailed description in the text or in the figure legend.
